

# Comparative transcriptome analysis of roots, stems, and leaves of *Pueraria lobata* (Willd.) Ohwi: identification of genes involved in isoflavonoid biosynthesis

Chenkai Wang[1,2,3], Nenggui Xu[1] and Shuai Cui[1,3]

[1] South China Research Center for Acupuncture and Moxibustion, Medical College of Acupuncture Moxibustion and Rehabilitation, Guangzhou University of Chinese Medicine, Guangzhou, China
[2] Anhui University, Hefei, China
[3] Anhui University of Chinese Medicine, Hefei, China

## ABSTRACT

**Background**. *Pueraria lobata* (Willd.) Ohwi is a valuable herb used in traditional Chinese medicine. Isoflavonoids are the major bioactive compounds in *P. lobata*, namely puerarin, daidzin, glycitin, genistin, daidzein, and glycitein, which have pharmacological properties of anti-cardiovascular, anti-hypertension, anti-inflammatory, and anti-arrhythmic.

**Methods**. To characterize the corresponding genes of the compounds in the isoflavonoid pathway, RNA sequencing (RNA-Seq) analyses of roots, stems, and leaves of *P. lobata* were carried out on the BGISEQ-500 sequencing platform.

**Results**. We identified 140,905 unigenes in total, of which 109,687 were annotated in public databases, after assembling the transcripts from all three tissues. Multiple genes encoding key enzymes, such as IF7GT and transcription factors, associated with isoflavonoid biosynthesis were identified and then further analyzed. Quantitative real-time PCR (qRT-PCR) results of some genes encoding key enzymes were consistent with our RNA-Seq analysis. Differentially expressed genes (DEGs) were determined by analyzing the expression profiles of roots compared with other tissues (leaves and stems). This analysis revealed numerous DEGs that were either uniquely expressed or up-regulated in the roots. Finally, quantitative analyses of isoflavonoid metabolites occurring in the three *P. lobata* tissue types were done via high-performance liquid-chromatography and tandem mass spectrometry methodology (HPLC-MS/MS). Our comprehensive transcriptome investigation substantially expands the genomic resources of *P. lobata* and provides valuable knowledge on both gene expression regulation and promising candidate genes that are involved in plant isoflavonoid pathways.

Corresponding authors
Nenggui Xu, ngxu8018@163.com
Shuai Cui, eferson@sina.com

## INTRODUCTION

The species *Pueraria lobata* (Willd.) Ohwi (*P. lobata*), commonly known as 'Kudzu', is a medical and edible plant in China that is widely distributed in temperate regions of China, Korea, Japan, and India (*Chen, Yu & Shi, 2018*; *Jin et al., 2012*). The dried root of *P. lobata* accumulates abundant isoflavonoids including puerarin, daidzein, daidzin, and genistein (*He et al., 2011*; *Wong et al., 2011*), and it has pharmacological activity for treating alcoholism, antioxidant, and other illnesses (*Carai et al., 2000*; *Miyazawa et al., 2001*). Puerarin is considered to be the main active isoflavonoid of *P. Radix* and *P. lobata* (*Du et al., 2010*; *Ohshima et al., 1988*; *Wang et al., 2017*), which reportedly has bioactivity against cardiovascular disease (*Pan et al., 2011*), vascular hypertension (*Han et al., 2015*) and it can improve insulin sensitivity (*Meezan et al., 2005*). Isoflavonoid synthesis is a branch of the flavonoid pathway. In the upstream pathway, the conversion of phenylalanine to liquiritigenin and naringenin happens through successive actions of phenylalanine ammonia-lyase (PAL), trans-cinnamate 4-monooxygenase (C4H), 4-coumarate-CoA ligase (4CL), chalcone synthase (CHS), and chalcone isomerase (CHI) (*Deavours & Dixon, 2005*; *Vogt, 2010*). In the downstream pathway, isoflavonoids are synthesized from liquiritigenin or naringenin under the hydroxylation, methylation, and glycosylation by 2-hydroxyisoflavanone synthase (IFS), 2-hydroxyisoflavanone dehydratase (HIDH), methyltransferases and glycosyltransferases, respectively (*Li et al., 2016*; *Wang et al., 2017*).

However, the study of molecular biology of *P. lobata* is relatively limited and only 6,365 ESTs sequences are available from the NCBI database (*He et al., 2011*). The lack of reference genome information of *P. lobata* plant hinders the further studies on the underlying genes involved in essential biological processes related to isoflavonoid biosynthesis. Additionally, TFs potentially involved in isoflavonoid biosynthesis pathway in *P. lobata* have not been reported except the research on isoflavonoid-related TFs in *P. thomsonii* (*He et al., 2019*).

RNA sequencing (RNA-Seq) analysis has been shown to be an efficient methodology for functional gene discovery and the identification of secondary metabolite pathways in those non-model plant species lacking known genomic sequences (*Garg & Jain, 2013*). Therefore, RNA-Seq is a powerful tool for the capturing of both coding and non-coding sequences, the mining functional genes and the quantification of gene expression levels (*Ozsolak & Milos, 2011*; *Wang, Gerstein & Snyder, 2009*; *Zhang, He & Cai, 2018*). At present, this approach was successfully used for transcriptome sequencing of some other medicinal plants, such as *Zanthoxylum planispinum* (*Kim et al., 2019*), *Trachyspermum ammi* (*Soltani Howyzeh et al., 2018*), *Picrorhiza kurroa* (*Vashisht et al., 2016*), and *Salvia miltiorrhiza* (*Chang et al., 2019*).

In this study, we performed a deep transcriptome analysis of three different tissues (roots, stems, and leaves) of *P. lobata* plants and identified the structural genes and transcription factors (TFs) potentially involved in isoflavonoid biosynthesis. This study forms the basis for further exploration of molecular mechanisms of isoflavonoid biosynthesis in *P. lobata*.

## MATERIALS & METHODS

### Plant material and RNA extraction

The materials (roots, leaves, and stems) of *P. lobata* were collected, in August 2018, from the town of Huaiyuan, Tongbai County, and Nanyang City, all located in Henan Province, China, with verbal permission of the Manager Hongchao Shi (Guangzhou Baiyunshan Chen Liji Pharmaceutical Factory Co., Ltd.), and authenticated by Professor Dequn Wang (Anhui University of Chinese Medicine). Separated tissues of three individual plants were immediately frozen in liquid nitrogen and stored at −80 °C until their respective extraction of RNA. Total RNA was extracted from each of the three tissues (roots, leaves, stems) of *P. lobata* plants in three replicates, by using the Spectrum Plant RNA Kit (Aidlab Biotech, Beijing, China), and following the manufacturer's instructions. The quality and quantity of total RNA were respectively analyzed using the NanoDrop spectrophotometer and Agilent 2100 Bioanalyzer (Thermo Fisher Scientific,Waltham, MA, USA) (Table S1).

### Isoflavonoid extraction and HPLC-MS/MS determination

For the isoflavonoid quantifications, 0.5 g of dried *P. lobata* tissues (roots, leaves, or stems) were directly extracted with 40 ml of methanol by ultrasonication for 20 min and then centrifuged at 10,000 rpm for 10 min. The supernatant was obtained for the following HPLC-MS/ MS analysis. Puerarin, daidzin, glycitin, genistin, genistein, daidzein, glycitein, and formononetin standards were obtained from Shanghai Yuanye Biological Technology Co., Ltd (Shanghai, China); for each, a standard calibration curve was established at concentrations of 0.1 mg/L, 0.2 mg/L, 0.5 mg/L, 1 mg/L, and 2 mg/L, respectively (Fig. S1). The HPLC-MS/MS system (Acquity, Waters, Milford, MA, USA) was used to analyze isoflavonoid metabolomics. Chromatographic separations of all samples were performed on Agilent ZORBAX Eclipse Plus C18 (dimensions: 2.1 mm ×100 mm, 1.8 μm particle size), using a gradient mixture of water (solvent A) and acetonitrile (solvent B) as the mobile phase. The column's temperature was controlled at 35 °C and the flow rate set to 0.3 mL/min. The gradient conditions were optimized as follows: 0∼30 min, 5% ∼25% B; 30∼40 min, 25%∼45% B; 40.01∼45 min, 5% B. The sample injection volume was 10 μL. Mass spectrometry was carried out with an Agilent LC/MS/MS 6460 mass spectrometer (Agilent Technologies Inc., Santa Clara, CA, USA) in negative ion mode with an ESI source. The optimized conditions for the mass spectrometer were as follows: ESI source voltage of 3.5 kV; nebulization with nitrogen, at 40 psi; nebulizer flow at 8.0 L/min, and a temperature of 350 °C. The scan range of mass spectra was set to 50–2,000 m/z, and ultra-high pure helium (He) served as the collision gas. The optimized MS parameters of isoflavonoid compounds can be found in Table S2.

### Library construction and RNA sequencing

The total RNA isolated from each plant tissue sample was applied to RNA-Seq library preparation, by following the protocol described in *Zhu et al. (2018)*. Briefly, total RNA was first treated with DNase I (TaKaRa, China) to remove all traces of genomic DNA; next, the mRNA molecules were enriched using oligo(dT)-attached magnetic beads and fragmented into smaller pieces by applying divalent cations, followed by their first-strand

and second-strand cDNA synthesis. Then, these cDNA fragments were subjected to end-repair and 3′adenylation and ligated to adapters, after which they were PCR-amplified to generate the final cDNA library. The single-stranded circular DNA was formatted as the final cDNA library for evaluation in the Agilent 2100 Bioanalyzer (ABI, New York, NY, USA), and later sequencing on the BGISEQ-500 platform (Beijing Genomics Institute, Wuhan, China).

## Transcriptome assembly

High-quality reads were obtained by discarding any reads with adaptors, unknown nucleotides (>5%) and those of low quality using the software tool SOAPnuke (*Chen et al., 2017*) (v1.4.0; parameters: -l 5 -q 0.5 -n 0.1) and Trimmomatic (v0.36; parameters: ILLUMINACLIP:2:30:10 LEADING:3 TRAILING: 3 SLIDINGWINDOW: 4:15 MINLEN:50). Since a reference genome for *P. lobata* was lacking, the high-quality reads obtained were assembled *de novo* by using Trinity (*Grabherr et al., 2013*) (v2.8.4). The assembled transcripts were then grouped using the TGI clustering tool (TGICL) (*Pertea et al., 2003*), discarding redundant sequences and retaining the non-redundant sequences, the latter termed unigenes.

## Functional annotation and differential expression analysis

Functional annotation was done by comparing homologous sequences against the UniProt databases using BLASTX (*Yang, Jiang & Zhang, 2014*) with an $E$-value threshold of $1.0e-5$. Also searched in this way were the databases of NR (non-redundant protein sequence), NT (nucleotide), KOG (clusters of euKaryotic Orthologous Groups), KEGG (Kyoto Encyclopedia of Genes and Genome), and SwissProt (a manually annotated and reviewed protein sequence database). The GO (Gene Ontology) annotation was carried out using Blast2GO (v2.5.0) (*Conesa & Götz, 2008*), and the sequences searched for in the Pfam (protein families) with the Hmmscan tool (v3.0) set to its default parameters.

To identify differentially expressed mRNAs in roots compared to leaves or stems, clean transcriptome reads of each sample were mapped onto the genome sequence of *P. lobata* using Bowtie2 (v2.2.5) (*Langmead & Salzberg, 2012*). The expression levels of unigenes were calculated using the RSEM (v1.2.8) (*Li & Dewey, 2011*) software program; those unigenes with a fold-change (FC) $\geq 2.00$ and FDR (false discovery rate) $\leq 0.001$ were designated as DEGs (differentially expressed genes) by the PoissonDis method (*Dembélé & Kastner, 2014*; *Kim & Van de Wiel, 2008*).

Multiple sequence alignment was conducted with Clustalx v2.0 software. Conserved domains in the amino acid sequence of *P. lobata* IF7GT were identified via the Conserved Domains Database (http://www.ncbi.nlm.nih.gov/Structure/cdd/wrpsb.cgi/). The secondary structure of *P. lobata* IF7GT was predicted using the SWISS-MODEL (*Bertoni et al., 2017*) and ESPript v3.0 software (*Robert & Gouet, 2014*).

## Analysis of transcription factor (TF)

The respective ORF of unigenes in the transcriptome of *P. lobata* root, leaf, and stem tissues was detected by Getorf software (*Rice, Longden & Bleasby, 2000*). Each was then mapped

to the plant TF database (PlantTFDB) by Hmmsearch (*Mistry et al., 2013*) using BLASTX (*E*-value $\leq$ 1e−5).

### Quantitative real-time PCR (qRT-PCR) analysis

To validate the transcriptome data of *P. lobata*, a qRT-PCR was carried out on the PIKOREAL 96 PCR System (Thermo Scientific, Waltham, MA, USA) with the Novostart SYBR qPCR SuperMix Plus kit (Novoprotein, Shanghai, China). Primer pairs were designed to amplify the actin gene and nine unigenes involved in isoflavonoid biosynthesis using Primer Premier (v5.0) (Table S3). All reactions were prepared in a final volume of 10 µl (containing 1 µl of cDNA, 1 µl of each specific primer, 5 µL of 2× SYBR Green mixture, and 2 µL of RNase-free water). The qRT-PCR was run under these conditions: 1 min at 95 °C, 40 cycles of 95 °C for 20 s, and 60 °C for 1 min. Relative expression levels were normalized to those of actin (CL10129.Contig5) mRNA, by applying the $2^{-\Delta\Delta Ct}$ method (*Livak & Schmittgen, 2001*). Every target gene was run in three biological replicates.

# RESULTS

### Determination of the content of isoflavonoid components in different *P. lobata* tissues

The quantitative analysis of isoflavonoid metabolites of different *P. lobata* tissues was done using HPLC-MS/MS. The resulting mass spectra indicated the presence of six isoflavonoid metabolites in *P. lobata* roots (puerarin, 2.649 mg/g; daidzin, 1.105 mg/g; glycitin, 0.196 mg/g; genistin, 0.0834 mg/g; daidzein, 0.052 mg/g; formononetin, 0.004 mg/g, for which mg/g is the fresh weight of roots), three metabolites in leaves (puerarin, 0.011 mg/g; genistin, 0.002 mg/g; genistein, 0.001 mg/g, for which mg/g is the fresh weight of leaves), and four metabolites in stems (puerarin, 0.005 mg/g; genistin, 0.005 mg/g; genistein, 0.001 mg/g; daidzein, 0.001 mg/g, for which mg/g is the fresh weight of stems), according to their respective retention time, molecular ion peak, and ESI-MS data (Fig. 1, Fig. S2). The content of isoflavonoid components was the highest in the roots (at 4.089 mg/g), with substantially lower amounts in the leaves (at 0.014 mg/g) and stems (0.013 mg/g); in this respect, the difference between roots and other tissue types was statistically significant.

### Illumina sequencing and de novo assembly of the *P. lobata* transcriptome

The RNA-Seq analysis of *P. lobate* roots, leaves, and stems generated 31.63 Gb of high-quality reads, characterized by a Q30-values $\geq$ 89.25% (Table S4). After assembling these high-quality reads per tissue type, their full-length transcripts were sequentially reconstructed. After selecting the longest transcript of each, a total of 140,905 unigenes were obtained (using software TGICL). These had a mean length of 1,083 bp and an N50 value of 1,883 bp; moreover, 38.47% (54,209) and 26.13% (36,821) of these unigenes exceeded 1,000 bp and 1,500 bp in length, respectively (Fig. S3; Table S5).

### Functional annotation and expression overview of unigenes

Of the 140,905 unigenes, we were able to map 70.60%, 58.04%, 55.30%, 55.03%, 53.67%, 49.62%, and 47.29% of them to the NR, NT, GO, KOG, KEGG, SwissProt, and Pfam

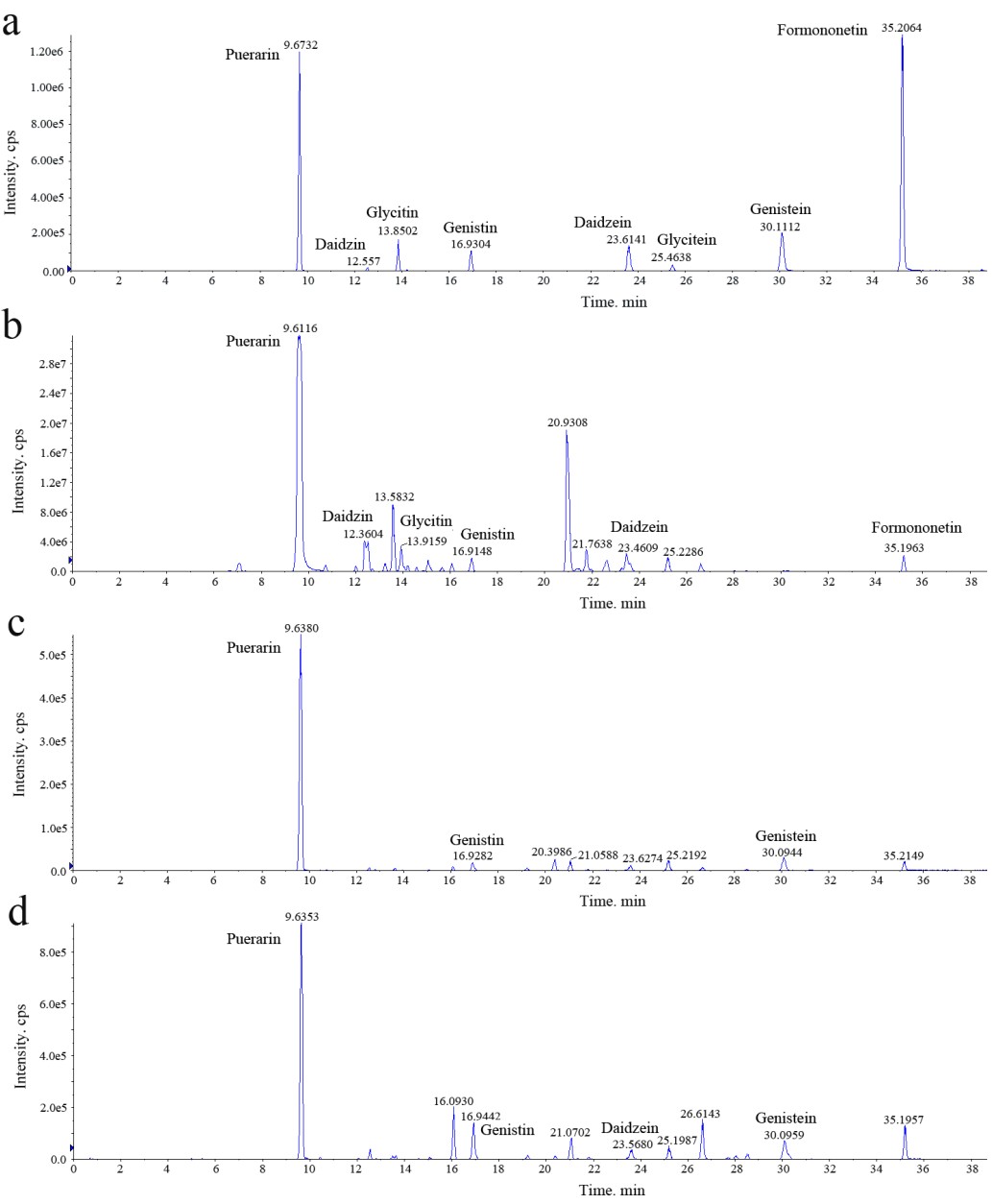

**Figure 1** **Representative chromatograms of the isoflavonoid analyzed in *Pueraria lobata* three tissues.**
(A) Standard sample solution; (B), (C) and (D) are respectively the root, leaf, and stem tissues. On the *x*-axis is the retention time; on the *y*-axis are the intensity values.

databases, respectively. Thus, a total of 109,687 (77.84%) unigenes were mapped to at least one public database, while 26.61% unigenes (37,498) were co-annotated in all seven databases (Table 1). A species distribution analysis showed that *P. lobata* unigenes had the highest homology to *Glycine soja* sequences (29,597 unigenes; 29.75%), followed by *Quercus suber* (18,164 unigenes; 18.26%) and *Glycine max* (16,822 unigenes; 16.91%) (Fig. S4).

**Table 1 *Pueraria lobata* unigenes annotated by seven public databases.**

| Database | Number annotated | Annotated unigene ratio (%) |
|---|---|---|
| NR | 99,480 | 70.60 |
| NT | 81,787 | 58.04 |
| GO | 77,920 | 55.30 |
| KOG | 77,533 | 55.03 |
| KEGG | 75,628 | 53.67 |
| Swissprot | 69,916 | 49.62 |
| Pfam | 66,629 | 47.29 |
| Intersection | 37,498 | 26.61 |
| Overall | 109,687 | 77.84 |

The GO database mapped 77,920 unigenes to "biological process", "cellular component" and "molecular function", with every unigene categorized to at least one GO term. Additionally, the abundant genes could be divided into "cellular process" (35,585 unigenes [45.67%]) and "metabolic process" (31,031 unigenes [39.82%]) terms under "biological process"; under the "molecular function", the "binding" (42,267 unigenes [54.24%]), and "catalytic activity" (40,388 unigenes [51.83%]) were the most abundant GO terms characterizing the unigenes (Fig. S5).

Unigenes with fragments per kilobase of transcript per million (FPKM) values were counted in different tissues of *P. lobata* plants. We found 42,591, 49,524, and 48,806 unigenes in the leaf, stem, and root tissues with an FPKM $\leq 1$, which corresponded to low-level expression; correspondingly, there were 29,780, 43,555, and 34,229 unigenes with an FPKM $= 1$–10, indicative of medium-level expression, and 9,999, 10,860, and 10,063 unigenes with an FPKM $\geq 10$ that exhibited high-level expression (Fig. 2A). Overall, the expression level was greatest in stems compared with the roots and leaves (Fig. 2B).

## Identification of structural genes involved in isoflavonoid biosynthesis

To identify the most significant biological pathways in *P. lobata* plants, functional annotations using the KEGG database mapped 75,628 unigenes to 19 subcategories (137 pathways). Of the latter, 11 were related to metabolism and most unigenes were assigned to "carbohydrate metabolism" (7,174 unigenes), followed by "amino acid metabolism" (3,888 unigenes), "lipid metabolism" (3,301 unigenes), "energy metabolism" (2,574 unigenes), "biosynthesis of other secondary metabolites" (2,567 unigenes) (Fig. 3A; Fig. S6, Tables S6–S8). Within the biosynthesis of the other set of secondary metabolites, the largest number of unigenes were found involved in the "phenylpropanoid biosynthesis" (1,590 unigenes), "flavonoid biosynthesis", "isoquinoline alkaloid biosynthesis", "tropane, piperidine, and pyridine alkaloid biosynthesis", and "isoflavonoid biosynthesis" pathways (Fig. 3B; Figs. S7 and S8).

Additionally, to enhance our understanding of isoflavonoid biosynthesis, we annotated 1,590 and 314 unigenes involved in the phenylpropanoid (ko00940) and flavonoid

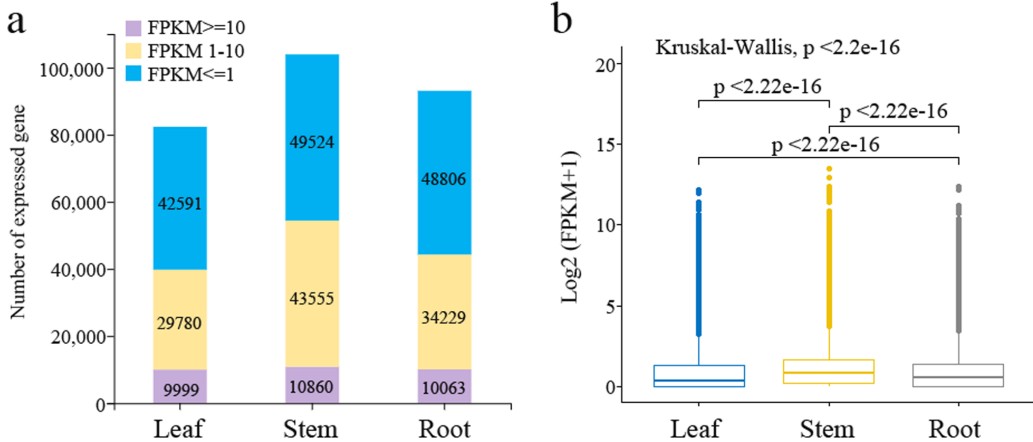

**Figure 2** Analysis of expression profiles in *Pueraria lobata* three tissues (roots, leaves, and stems). (A) Distribution of number of unigenes having different expression levels in tissues. (B) Expressed unigenes in tissues depicted as boxplots. Statistical comparisons of the medians were made with the Kruskal–Wallis nonparametric test.

biosynthesis (ko00941) pathways, respectively. A total of 94 unigenes that encoded key enzymes were identified under "isoflavonoid biosynthesis" (ko00900): these were phenylalanine ammonia-lyase (PAL; 8 unigenes), 4-coumarate-CoA ligase (4CL; 17 unigenes), trans-cinnamate 4-monooxygenase (C4H; 9 unigenes), chalcone synthase (CHS; 7 unigenes), chalcone isomerase (CHI; 19 unigenes), 2-hydroxyisoflavanone synthase (IFS2; 11 unigenes), flavonoid 6-hydroxylase (F6H; 1 unigenes), isoflavone 4′-O-methyltransferase (IOMT; 4 unigenes), 2-hydroxyisoflavanone dehydratase (HIDH; 11 unigenes) and isoflavone 7-O-glucosyltransferase (IF7GT; 7 unigenes) (Table 2; Table S9). A heatmap was used to visualize patterns in the expression levels of these enzyme-encoding unigenes involved in isoflavonoid biosynthesis pathways (Fig. 4). Most of the unigenes encoding PAL, C4H, CHS, F6H, and IF7GT were expressed at higher levels in roots, while some unigenes encoding IOMT and HIDH were expressed more in leaves. Unigenes encoding 4CL, CHI, and IFS2 were all highly expressed in stems (Fig. 4).

In our assembled transcriptome for *P. lobata*, a total of 298 unigenes encoding the UDP glycosyltransferases (UGTs) were identified, which included 137 full-length open reading frames (ORFs) of the unigenes (Table S10). Furthermore, seven unigenes (CL4436–1, CL4436–4, CL4436–5, CL4436–6, CL4436–8, CL4436–9, Un–54862) were predicted to encode the isoflavone 7-O-glucosyltransferase (IF7GT) known to be involved in isoflavonoid biosynthesis, whose expression among tissues was greatest in the roots. The amino acid sequence of IF7GT in *P. lobata* revealed a highly conserved region within the C-terminal domain—known as a plant secondary product glycosyltransferase (GT) consensus sequence (called PSPG)—involved in secondary metabolism when compared with the other plant UGTs of *G. max* (NP_001304440.2; 93.53%), *G. soja* (XP_028207064.1; 93.28%), and *P. montana* var.lobata (AMQ26113.1; 90.55%).

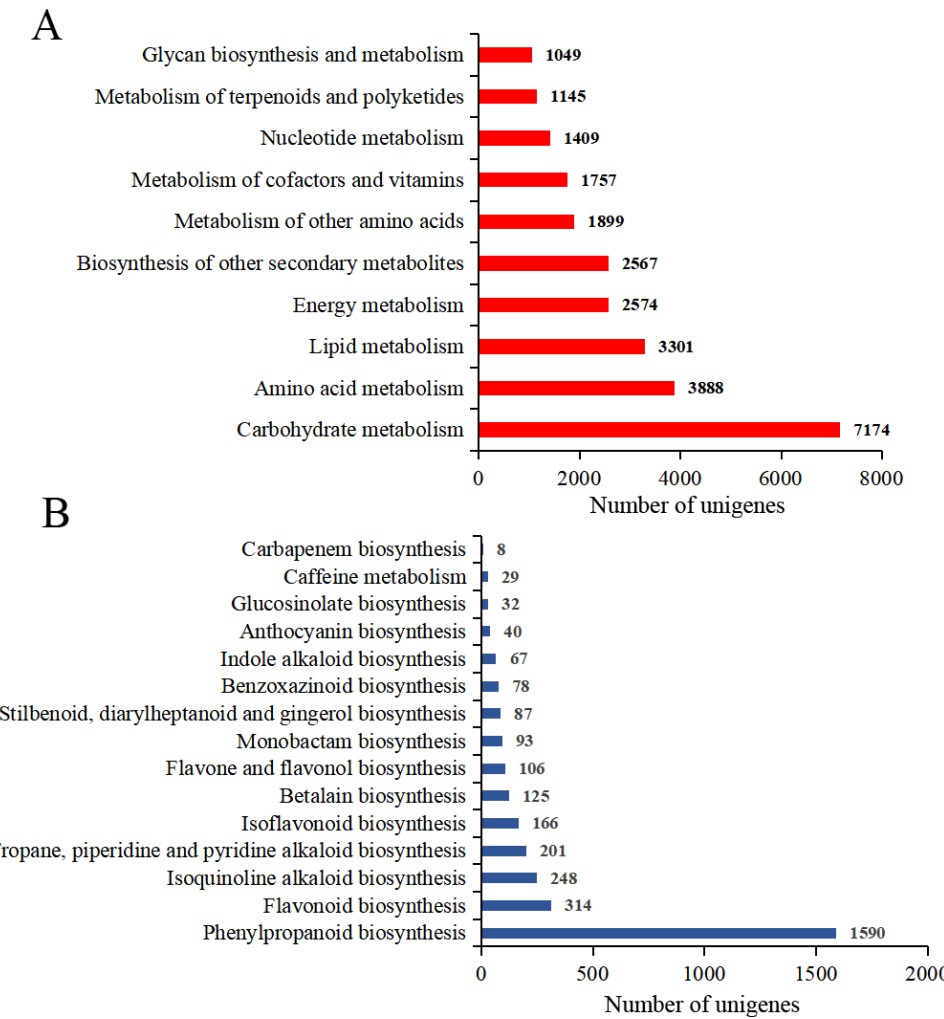

**Figure 3** **KEGG annotation of *Pueraria lobata* unigenes.** Number of unigenes that participated in the metabolites' classification (A) and other secondary metabolites' classification (B).

The secondary structure of IF7GT contained 12 β-sheets and 14 α-helices, with the regions of β-sheet 10 to α-helix 14 potentially involved in binding the protein to the UDP moiety of sugar (depicted by the black box in Fig. 5).

## Validation of RNA-Seq data using qRT-PCR

To verify whether the differential gene expression found among different *P. lobata* tissues by the FPKM analysis was accurate, nine unigenes participating in isoflavonoid biosynthesis were chosen for a qRT-PCR analysis. The expression levels of the six unigenes including C4H (CL2444.Contig2), 4CL (CL1520.Contig3), CHS (CL3338.Contig1), IFS2 (CL2625.Contig2), HIDH (CL10538.Contig1), and IF7GT (Unigene54862) were all consistent between the qRT-PCR analysis and RNA-Seq data. Furthermore, the qRT-PCR results corroborated that the expression levels of these selected genes were highest in root tissue (Fig. 6).
**Table 2   Summary of isoflavonoid biosynthesis unigenes in three *Pueraria lobata* tissues.**

| Enzyme name | EC number | Unigene number | No. in roots | No. in stems | No. in leaves |
|---|---|---|---|---|---|
| PAL | 4.3.1.24 | 8 | 8 | 8 | 8 |
| C4H | 1.14.13.11 | 9 | 9 | 8 | 7 |
| 4CL | 6.2.1.12 | 17 | 16 | 16 | 15 |
| CHS | 2.3.1.74 | 7 | 7 | 7 | 7 |
| CHI | 5.5.1.6 | 19 | 17 | 19 | 18 |
| IFS2 | 1.14.13.136 | 11 | 8 | 11 | 9 |
| F6H | 1.14.13.- | 1 | 1 | 0 | 1 |
| IOMT | 2.1.1.212 | 4 | 4 | 3 | 4 |
| HIDH | 4.2.1.105 | 11 | 10 | 10 | 11 |
| IF7GT | 2.4.1.170 | 7 | 7 | 7 | 6 |

**Notes.**
EC, Enzyme Commission.

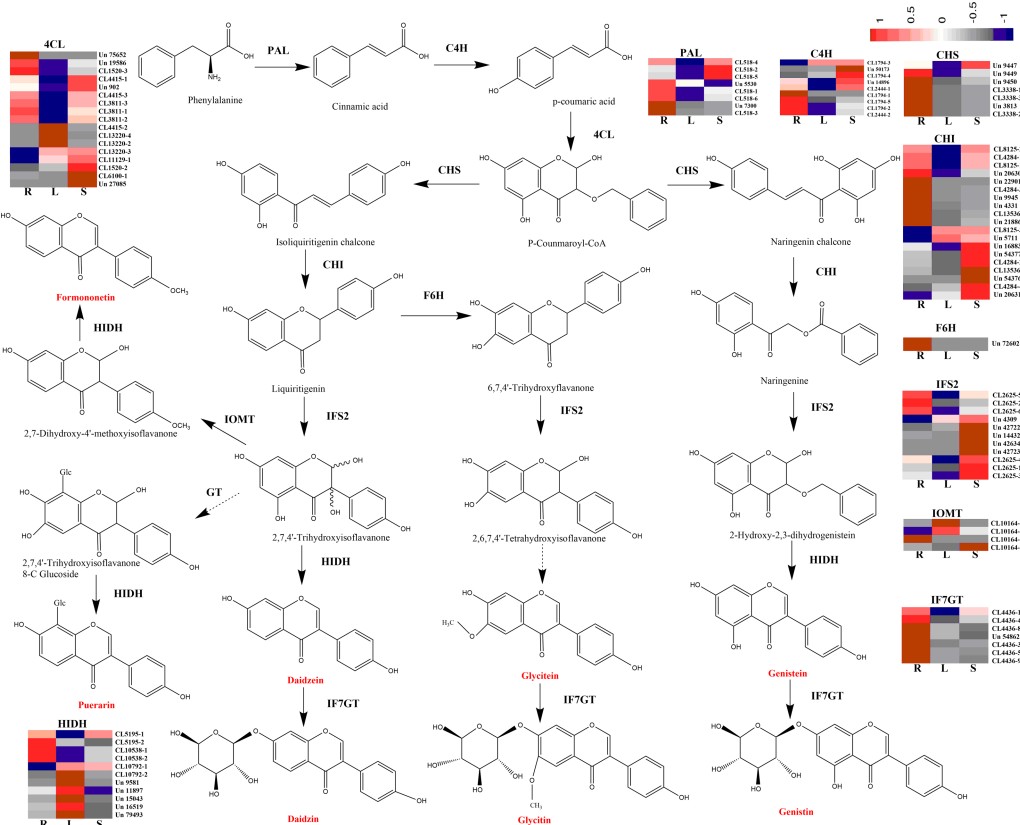

**Figure 4   Predicted pathways for isoflavonoid biosynthesis in *Pueraria lobata*.** The heatmap shows the expression levels of unigenes encoding enzymes of the key metabolic entry point for the formation of all isoflavonoids. R, roots; L, leaves; S, stems. "Un" and "CL" represents a cluster of unigenes and transcripts, respectively. Red and blue indicate the high- and low-expression levels, respectively. Different constituents of isoflavonoids are marked in red.

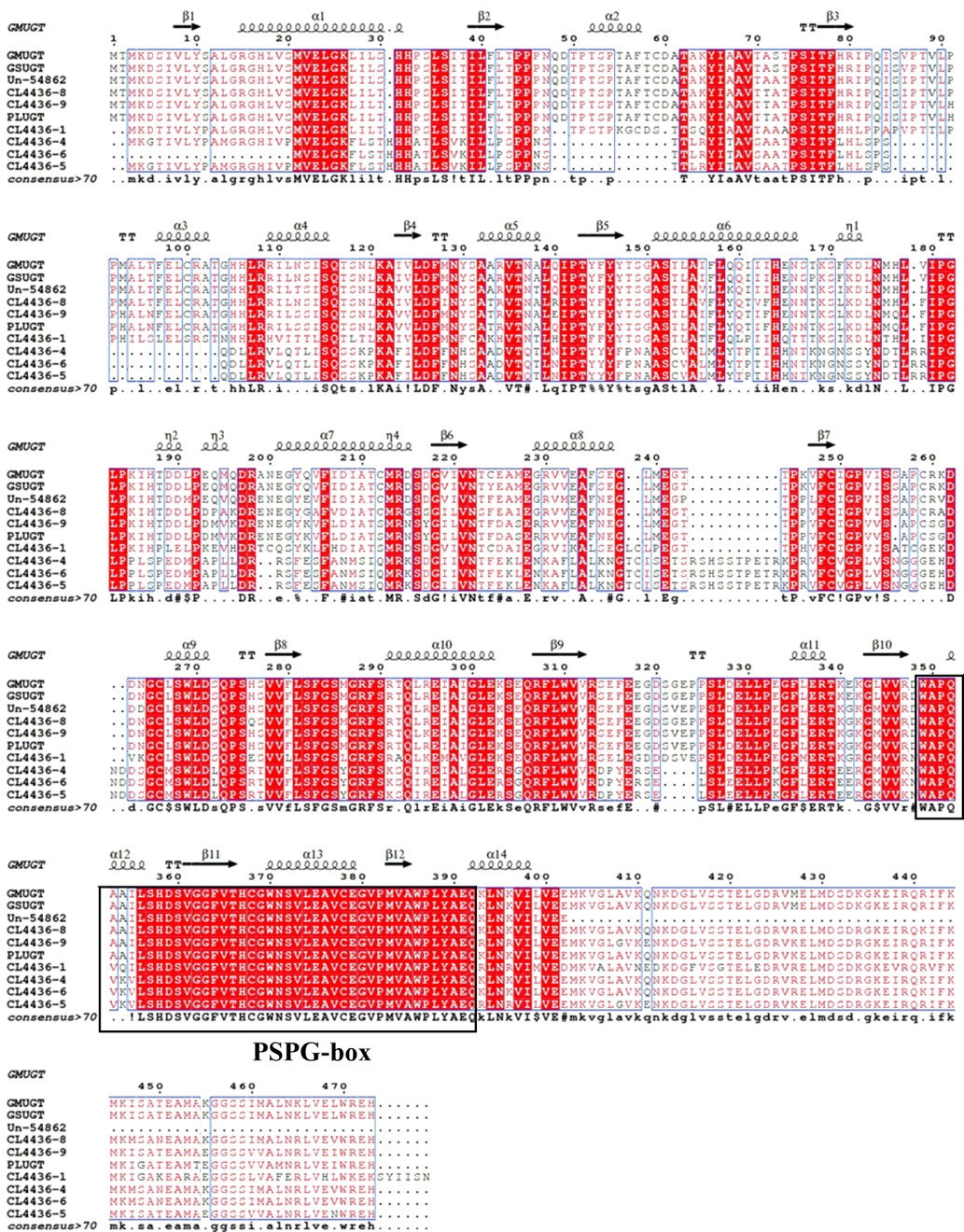

**Figure 5  Multiple sequence alignment and secondary structure of *Pueraria lobata* IF7GTs with other representative UGTs.** The red-highlighted sections and red letters on a white background respectively represent identical and similar amino acids. A conserved PSPG motif (potential UDP-binding domain) is marked in black box. GMUGT, isoflavone 7-O-glucosyltransferase UGT4 in *Glycine max* (NP_001304440.2); GSUGT, UDP-glycosyltransferase 1 in *Glycine soja* (XP_028207064.1); isoflavone 7-O-glucosyltransferase in *Pueraria montana* var. *lobata* (AMQ26113.1).

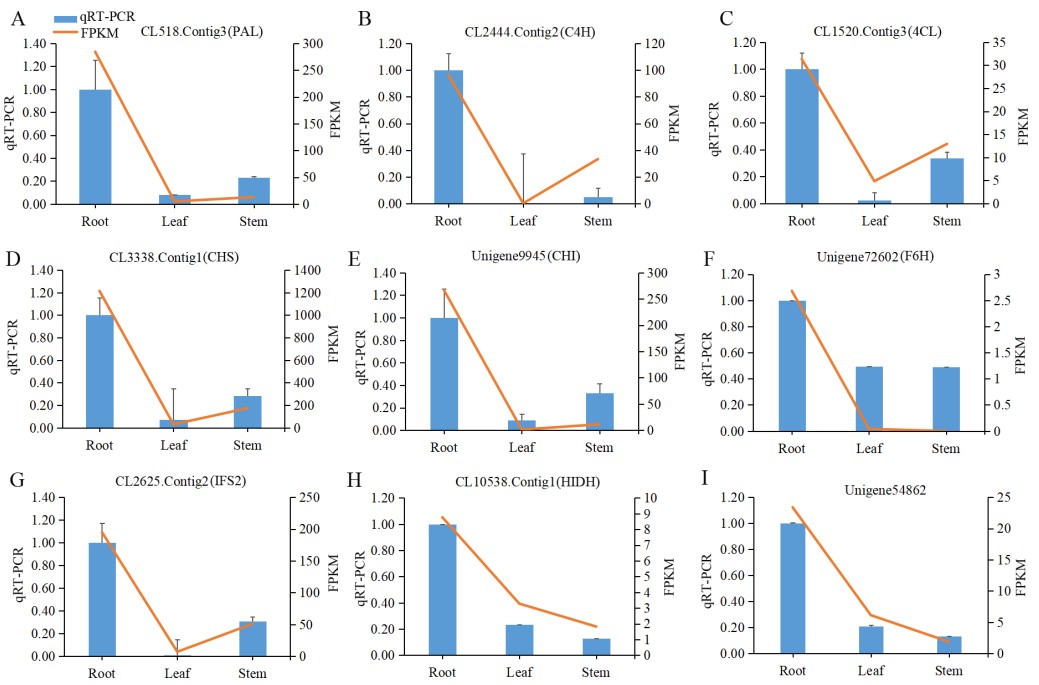

**Figure 6  The qRT-PCR analysis of nine candidate unigenes encoding enzymes involved in isoflavonoid biosynthesis in different tissues of *Pueraria lobata*.** Relative expression of CL518.Contig3 (PAL), CL2444.Contig2 (C4H), CL1520.Contig3 (4CL), CL3338.Contig1 (CHS), Unigene9945 (CHI), Unigene72602 (F6H), CL2625.Contig2 (IFS2), CL10538.Contig1 (HIDH), Unigene54862 (IF7GT) was normalized with respect to the actin gene (CL10129.Contig5). Blue bars indicate the qRT-PCR results, and red lines show the FPKM values identified via the RNA-Seq analysis. Data is shown as the mean ± standard error of three replicates. The left *y*-axis is the relative expression level of unigenes obtained by qRT-PCR, and the right *y*-axis denotes the FPKM values in the RNA-Seq data.

## Identification of transcription factors (TFs) involved in the biosynthesis of isoflavonoid and other secondary metabolites

The mechanism regulating isoflavonoid metabolic processes is considered to operate primarily at the transcription level, so TF families likely play key roles in this by specifically binding to cis-regulatory elements in the promoter regions of involved genes. A total of 2,985 unigenes were identified as putative TFs in the *P. lobata* transcriptome when using Hmmsearch (v3.0) to search the plant transcription factor database (PlantTFDB). These included 905 and 474 unigenes up-regulated in roots compared with stems and leaves, respectively (Table 3; Table S11). Most of these TF-encoding unigenes were annotated to MYB (375 unigenes), bHLH (221 unigenes), WRKY (208 unigenes), AP2-EREBP (196 unigenes), NAC (170 unigenes), C3H (158 unigenes), and C2H2 (156 unigenes) families. Furthermore, we found that the unigenes encoding the MYB (sixunigenes), TCP (three unigenes), Trihelix (two unigenes), as well as BSD, C2H2, GeBP, WRKY, mTERF, and zf-HD (one unigene each) TFs were involved in phenylpropanoid biosynthesis, providing the precursors for isoflavonoid metabolism. We also uncovered a total of 14 unigenes encoding several TFs (AP2-EREBP, MYB, RWP-RK, and zf-HD) involved in "betalain
**Table 3  Type and number of transcription factor (TF) families encoded via the DEGs (differentially expressed genes) database of *Pueraria lobata*.**

| TF family | Number of unigenes | Up-regulated unigenes in root vs leaf | Up-regulated unigenes in root vs stem |
|---|---|---|---|
| MYB | 375 | 120 | 57 |
| bHLH | 221 | 84 | 44 |
| WRKY | 208 | 66 | 36 |
| AP2-EREBP | 196 | 85 | 39 |
| NAC | 170 | 44 | 32 |
| C3H | 158 | 48 | 16 |
| C2H2 | 156 | 48 | 39 |
| G2-like | 101 | 16 | 13 |
| GRAS | 98 | 42 | 11 |
| FAR1 | 93 | 20 | 11 |
| ARF | 81 | 31 | 15 |
| MADS | 75 | 19 | 12 |
| Trihelix | 75 | 13 | 8 |
| ABI3VP1 | 73 | 21 | 15 |
| C2C2-GATA | 56 | 14 | 10 |
| mTERF | 56 | 18 | 5 |
| C2C2-Dof | 55 | 28 | 4 |
| LOB | 45 | 22 | 10 |
| SBP | 45 | 15 | 7 |
| Tify | 45 | 4 | 8 |
| HSF | 44 | 11 | 9 |
| LIM | 44 | 8 | 8 |
| TCP | 38 | 4 | 1 |
| FHA | 37 | 8 | 3 |
| TIG | 34 | 9 | 4 |
| TUB | 34 | 10 | 4 |
| Zn-clus | 32 | 2 | 2 |
| Alfin-like | 31 | 6 | 1 |
| bZIP | 27 | 4 | 8 |
| BSD | 24 | 3 | 3 |
| RWP-RK | 14 | 4 | 0 |
| zf-HD | 13 | 1 | 0 |
| GeBP | 8 | 2 | 0 |
| Other | 223 | 75 | 39 |
| Total number | 2985 | 905 | 474 |

biosynthesis", "carotenoid biosynthesis", and "tropane, piperidine and pyridine alkaloid" secondary metabolites.

## Identification of DEGs and analysis of potential TFs related to isoflavonoid biosynthesis

Among the unigenes identified in three different tissues of *P. lobata*, 18,170 were expressed specifically in roots, while 62,297 were expressed in all three tissues (Fig. 7A). DEGs were detected among tissue types according to the FPKM values of unigenes (Fig. 7B). A total of 38,751 DEGs between the root and leaf transcriptome were detected, of which 17,064 were down-regulated and 21,687 up-regulated in roots compared with leaves. Comparing the unigene expression levels between roots and stems revealed 36,155 DEGs, of which 21,884 were down-regulated and 14,271 were up-regulated in roots compared with stems. Far less DEGs (19,790) were identified when roots were compared to both leaves and stems, with 9,440 of these down-regulated and 10,350 up-regulated.

In addition to the structural unigenes, 19,790 co-expression DEGs were further analyzed to look for other regulatory unigenes, such as TFs, that were potentially related to the isoflavonoid biosynthesis. We found 798 co-expression DEGs were predicted to be TF families and it is hypothesized that isoflavonoid-related TFs should have highly correlated expression levels with those of structural genes. Thus, the expression levels of 798 TFs were calculated using Pearson's correlation coefficients (r) ($r > 0.8$ or $r < -0.8$) with those of PAL (CL518.Contig3), C4H (CL2444.Contig2), 4CL (CL1520.Contig3), CHS (CL3338.Contig1), CHI (Unigene9945), F6H (Unigene72602), IFS2 (CL2625.Contig2), HIDH (CL10538.Contig1), and IF7GT (Unigene54862). In total, 608 TFs that were coexpressed with at least 1 of the 9 structural unigenes involved in isoflavonoid biosynthesis were screened (Table S12). Among them, we also identified 426, 16, 45, 17, 8, 15, 31, 45 and 5 TFs were negatively or positively correlated with the 9, 8, 7, 6, 5, 4, 3, 2, and 1 structural unigene, respectively (Table S12). The 608 TFs were classified into 49 families, and most of the unigenes were enriched to the MYB, bHLH, WRKY, AP2-EREBP and C2H2 TF families.

## Analysis of root-specific expression unigenes

A total of 10,350 up-regulated DEGs that exhibited root-specific expression, with values of log2 FC >1, were mapped to 54 subcategories in the three functional categories of GO database. Within the biological processes and molecular function categories, most genes were significantly enriched under "cellular process", "metabolic process", and "catalytic activity", and "binding"categories (Fig. 8A; Table S13). Within the KEGG database, these DEGs having root-specific expression were further annotated to 134 pathways. Among the latter, most DEGs were significantly enriched in "flavonoid biosynthesis" (Rich ratio = 0.283), "isoflavonoid biosynthesis" (Rich ratio = 0.259), "stilbenoid, diarylheptanoid and gingerol biosynthesis" (Rich ratio = 0.241), "phenylpropanoid biosynthesis" (Rich ratio = 0.181), and "lysine biosynthesis" (Rich ratio = 0.179) (Fig. 8B).

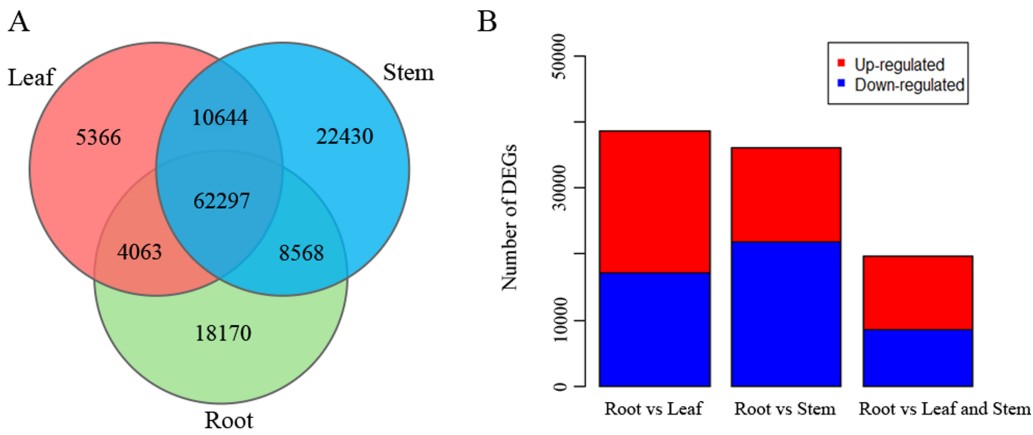

**Figure 7** **Expression of unigenes in *Pueraria lobata* root, stem, and leaf tissues.** (A) Number of unigenes expressed within and among the three tissues summarized in a Venn diagram. (B) The numbers of up- and down-regulated DEGs in root vs. leaf, root vs. stem, root vs. leaf and stem datasets.

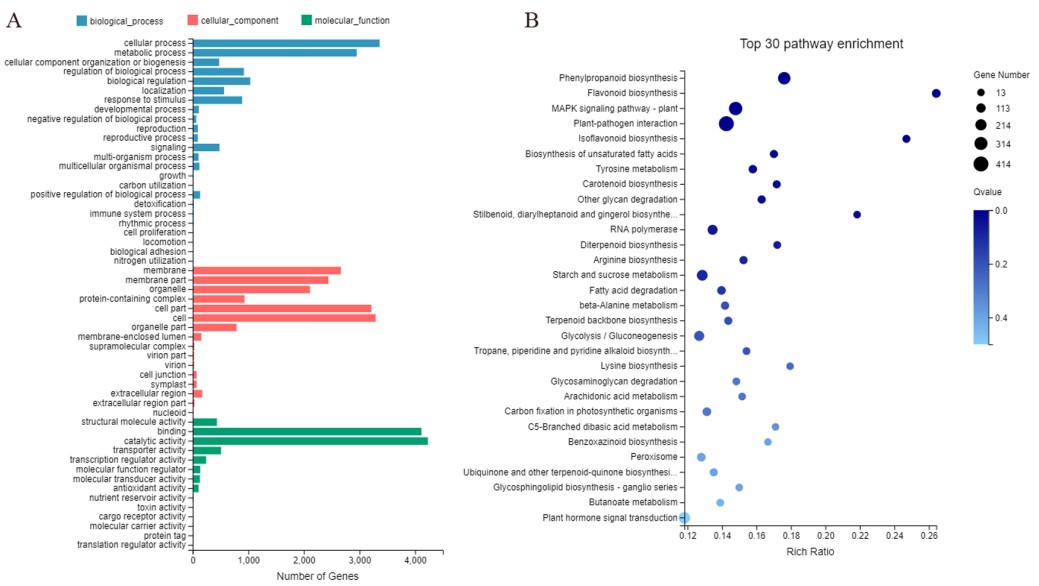

**Figure 8** **GO and KEGG enrichment analyses (A and B) of unigenes expressed specifically in the roots of *Pueraria lobata*.**

## DISCUSSION

The roots of *P. lobata*, which are the main plant part prescribed in traditional Chinese medicines to treat various diseases, produce predominantly isoflavone O- and C-glucosides. The three tissues from which the concentrations of isoflavonoids were detected using HPLC-MS/MS were also analyzed for deep transcriptomic data on the BGISEQ-500 system. Isoflavonoids clearly accumulate to a much higher level in roots than either leaves or stems, and the content of puerarin exceeds that of daidzin, glycitin, genistin, genistein,

daidzein, and formononetin (Fig. 1). Further, the concentration of puerarin in roots not only differs from leaves and stems, but it also varies among different root areas of the *P. lobata* plant (*Du et al., 2010*). This unequal distribution may arise from essential enzymes for the accumulation of puerarin being actively expressed in the roots with low or no expression in other plant tissue types. Thus, it is important to investigate these DEGs since this could advance the functional research of *P. lobata* plants with high contents of special isoflavonoids, to expand their usage. The analysis of the transcriptome dataset in our study revealed 140,905 unigenes, 109,687 of which were annotated, leaving 22.16% non-annotated, probably due to insufficient data on plant transcriptomes and genomes currently in the NCBI database. Nonetheless, the Nr annotations showed that 29,597 (29.75%) of the *P. lobata* unigenes had higher homology to the *G.soja* protein database, this having a higher similarity of sequences than with either *Q.suber* or *G.max*. This result suggested *P. lobata* is more closely related to the *G.soja* and so the latter's genome could be taken as a reference for future functional genomics research in *P. lobata*.

We performed RNA-Seq analyses of roots, stems, and leaves of *P. lobata* to facilitate the dissection of differentially expressed genes involved in the tissue-specific biosynthesis of isoflavonoid. This approach has been applied to identify the novel genes involved in the secondary metabolism pathways and analysis of the molecular mechanism of isoflavonoid biosynthesis (*Dastmalchi & Dhaubhadel, 2015*; *Suntichaikamolkul et al., 2019*; *Wang et al., 2018*). Comparative analysis of *P. lobata* roots and leaves and stems expression profiles exhibited 10,350 root-specific DEGs, and that these DEGs were significantly enriched in flavonoid biosynthesis and isoflavonoid biosynthesis (Fig. 8B). These root-specific DEGs may explain the molecular mechanism of the medicinal value of *P. lobata* roots.

Isoflavonoids are the major bioactive components in *P. lobata*, although the relationship between the expression of structural genes in isoflavonoid biosynthesis and the accumulation of isoflavonoids in *P. lobata* is unclear. In the transcriptome dataset of *P. lobata*, we identified 94 putative structural unigenes that were involved either upstream or downstream of the isoflavonoid biosynthesis pathway (Fig. 4). Our HPLC-MS/MS analysis confirmed that the content of isoflavonoids in *P. lobata* was the highest in roots compared with other tissues. Higher expression levels of some unigenes encoding F6H, HIDH, and IF7GT in roots compared with leaves and stems were consistent with the fact higher isoflavonoid accumulation occurred in *P. lobata* roots (Figs. 1, 4 and 6). Taken together, these results suggest overexpression of these genes could increase the accumulation of isoflavonoids in *P. lobata* plants. Previous studies have shown that overexpression of the HIDH gene increased the productivity of daidzein and genistein in *Lotus japonicas* roots (*Shimamura et al., 2007*), and that the expression level of a novel isoflavone 7-O-glucosyltransferase (PlUGT1) was correlated with the accumulation pattern of isoflavone glycosides in *P. lobata* roots (*Li et al., 2014*).

In plants, UDP glycosyltransferases (UGTs) utilize UDP-activated sugars as donors and transfer their sugar moiety to acceptor molecules, which play central roles in regulating isoflavonoid activities and detecting the diversity of their structures (*Li et al., 2007*; *Noguchi et al., 2007*). Within the *P. lobata* transcriptome, seven full-length IF7GT candidates were selected that had their highest expression levels in roots. Multiple sequence alignment

of the IF7GT amino acids revealed a highly conserved region in the PSPG-box (Fig. 5), which is similar to the original GT consensus sequence, with a short stretch of ~40 amino acids at the C-terminal part of the protein (*Bairoch, 1992*; *Vogt & Jones, 2000*). GTs are generally involved in plant secondary metabolism, but this depends on the inversion of the anomeric sugar from an α linkage in UDP-glucose to the β configuration in resultant glycoside (*Dmitri & Yu, 1999*). Two highly conserved sequences (WAPQA and HCGWNS) of IF7GT were located at β10, α12, β11, and α13, and they may correspond to the active site of GTs in *P. lobata*. Thus, our findings can assist further investigation into the structures of IF7GTs and the regulation of isoflavonoid activities.

In addition to the structural unigenes, we also analyzed TFs regulatory unigenes. TFs are sequence-specific DNA-binding proteins that can either induce or repress the accumulation of specific metabolites, quite effectively, by regulating the expression of target genes (*Butelli et al., 2008*; *Cutanda-Perez et al., 2009*; *Shelton et al., 2012*). For example, overexpression of the *LjMYB14* gene can enhance the expression of genes that encode PAL, C4H, and 4CL enzymes involved in the isoflavonoid biosynthesis of *Lotus japonicas* (*Shelton et al., 2012*). In soybean plant, over-expression of *GmMYB176* by activating CHS8 gene promoter affected isoflavone biosynthesis (*Yi et al., 2010*). Conversely, over-expression of *GmMYB39* in soybean was able to reduce the expression levels of PAL, C4H, CHS, and 4CL, yet it slightly increased that of IFS (*Liu et al., 2013*). Here, we found that the expression level of MYB39 (XM_003541813.2) was positively correlated with that of PAL, C4H, 4CL, CHS, CHI, F6H, IFS2, HIDH, and IF7GT in *P. lobata*. Additionally, we also identified TFs of bHLH, WRKY, AP2-EREBP, C2H2, NAC,G2-like and MADS that were potentially involved in isoflavonoid biosynthesis by regulating multiple structural unigenes (Table S12). Knowledge of these TFs existence could foster functional studies of transcriptional activation of key enzymes related to isoflavonoid synthesis in *P. lobata*.

## CONCLUSIONS

Overall, deep transcriptome analysis of root, leaf, and stem tissues of *P. lobata* plants were done for identifying candidate genes involved in this plant's biosynthesis of isoflavonoids. The results of this RNA-Seq analysis will facilitate further study of the molecular mechanisms of isoflavonoid biosynthesis in *P. lobata* and should allow us to analyze the functional genomics other pathways that are currently less well understood.

## ACKNOWLEDGEMENTS

We thank the Beijing Genomics Institute for assistance with the experiments.

### Funding
The authors received no funding for this work.

### Competing Interests
The authors declare there are no competing interests.

## Author Contributions

- Chenkai Wang performed the experiments, analyzed the data, prepared figures and/or tables, authored or reviewed drafts of the paper, and approved the final draft.
- Nenggui Xu conceived and designed the experiments, analyzed the data, prepared figures and/or tables, and approved the final draft.
- Shuai Cui conceived and designed the experiments, performed the experiments, prepared figures and/or tables, and approved the final draft.

## Field Study Permissions

The following information was supplied relating to field study approvals (i.e., approving body and any reference numbers):

Field experiments were approved by the verbal permission of the Manager Hongchao Shi (Guangzhou Baiyunshan Chen Liji Pharmaceutical Factory Co., Ltd.) (project number: Z44021058).

## Data Availability

Data is available at NCBI SRA: PRJNA630835, SRP261192.

## Supplemental Information

Supplemental information for this article can be found online at http://dx.doi.org/10.7717/peerj.10885#supplemental-information.

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
