# Peer review of "Comparative transcriptome analysis of roots, stems, and leaves of Pueraria lobata (Willd.) Ohwi: identification of genes involved in isoflavonoid biosynthesis"

_PeerJ, doi:10.7717/peerj.10885_

## Round 0.1 · original submission · Major Revisions

Your manuscript has been reviewed by two experts in the field. As you can see from their comments below, both of them raise substantial criticisms on it. Please read them carefully and revise the manuscript accordingly. Particularly, the second reviewer points out that the data availability information is missing. It is PeerJ's policy that all the data must be deposited in a public place. In addition, both reviewers point out the need of polishing your description and/or professional proofreading.

Reviewer 1 ·

Basic reporting

The manuscript needs professional editing, there are many grammatical errors.

Experimental design

If you have find some new isoflvones in P. lobata by HPLC-MS?

Validity of the findings

The discussion part should be strenghrened.

Additional comments

1、The introduction and discussion should be strengthened. I suggest the authors read and refer to paper mentioned the isofavonoid biosynthesis in P. lobata or Pueraria thomsonii Benth.
(1)Han R, Takahashi H, Nakamura M, Yoshimoto N, Suzuki H, Shibata D, Yamazaki M, Saito K: Transcriptomic landscape of Pueraria lobata demonstrates potential for phytochemical study. Front Plant Sci 2015, 6:426.
(2) He M, Yao Y, Li Y, Yang M, Wu B, Yu D: Comprehensive transcriptome analysis reveals genes potentially involved in isoflavone biosynthesis in Pueraria thomsonii Benth. PLoS One 2019, 14(6):e0217593.
(3)He X, Blount JW, Ge S, Tang Y, Dixon RA: A genomic approach to isoflavone biosynthesis in kudzu (Pueraria lobata). Planta 2011, 233(4):843-855.
(4)Wang X, Li S, Li J, Li C, Zhang Y: De novo transcriptome sequencing in Pueraria lobata to identify putative genes involved in isoflavones biosynthesis. Plant Cell Rep 2015, 34(5):733-743.
2biosynthesis.
3、 Co-expression analysis should be performed between TF and isofavonoid related structural genes to identify TF potentially involved in isofavonoid biosynthesis.
4、Line 150-155: In part of Quantitative real-time PCR (qRT-PCR) analysis, the gene ID is not necessary.
5、Line 78:the GPS information of the P. lobata should be added.

Reviewer 2 ·

Basic reporting

The paper by Wang et al. describes generation of transcriptome data from leaves, stems and roots of Pueraria lobata. They also check for presence of 8 isoflavonoids in these tissues.

1) The manuscript has some serious flaws. Most importantly, the sequencing data have not been deposited. They are not available as raw data, and no statement has been made in the manuscript that they will be deposited in a public database. I agree with the authors that the transcriptomes would be a valuable resource for gene mining, but not if it is not shared. Without depositing the sequencing data into a public database, this study contributes absolutely nothing to this scientific area.

2) The manuscript also appears to be prepared rather sloppy.
2.1) For one, the word "Isoflavonoid" is miss-spelled in the title and a few times in the article.
2.2) The reference list is a mess, with years, editions, etc missing from several references.
2.3) The writing is confusing, and especially the introduction confused me (lines 50-60) and would benefit from a re-writing. There are several places also in the other sections that would benefit from rewriting, but I will spend my time in going over these that could have been caught by the authors in proofreading.
2.4) lines 166 onwards: the amount of flavonoids is given as "mg/kg". Please specify per kg of what? Also, this is a strange unit, giving the amount as "mg/g fresh weight" if that is what it is, would be the norm.
2.5) lines 195-200: what tissues do these number correspond to?

3) The discussion should be improved to be a discussion, and not a repeat of the results.

4) The design of the figures is sub optimal.
4.1) Figure 2 does not work in its current form. the bars for leaf and stem are not visible. It is also not really needed, the three values are reported in the text and the figure adds no more value as it is now.
4.2) Figure 3a has a bad choice of color. It is already close to impossible for a non-color-blind person to read. Please consider use of colors that are distinguishable also for color blind people. This is valid also for the other figures.

Experimental design

The research question is well defined, relevant and meaningful. The methods are described with sufficient detail.

I have a few comments to the experimental part of the paper.
1) qPCR uses three technical replicates. These should be biological replicates, otherwise the error bars just represent the authors pipetting skills.

Validity of the findings

1) Without the sequencing data deposited, it is impossible to determine the validity of the findings from the transcriptome experiment.

There are a few other concerns as well.
2) Line 170: "total isoflavonoid content". This is not total, but a sum of the 8 ones that were quantified.

3) The validation of the RNA-seq data using qRT-PCR (lines 238-245 and figure 7)
The qPCR should use biological replicates instead of technical. The authors conclude that "expression levels ... were all consistent between" qPCR and RNA-Seq, but this is only true for 6 out of the 9 genes analyzed by qPCR.

4) The manuscript contains many occurrences of biased phrasing. "this study is important...", "this is the first...". Consider rewriting these instances into neutral sentences.

---

## Round 0.2 · Minor Revisions

Your revised manuscript has been reviewed by one of the original referees (unfortunately, the other did not respond to my invitation for re-review but I confirm that the manuscript has been revised appropriately on the points raised by the second reviewer). As you will see in the comments below, the reviewer admits that the revision has improved the manuscript significantly but also points out that some important issues remain/emerge. Particularly, the issue of the biological replicates is important. Please read the comments carefully and re-revise the manuscript accordingly.

Reviewer 2 ·

Basic reporting

The readability of the manuscript by Wang et al. has improved by English language editing.
Also, the sequencing data have been deposited in the NCBI Sequence Read Archive and can now be accessed.

The discussion section is improved somewhat, but I am still missing a discussion of the findings in relation to published work (for example by He et al., 2019, 2011; Wang et al., 2015 and Han et al., 2015).

However, there is still an issue with the qPCR data.

Comments to the Experimental design
1) qPCR uses three technical replicates. These should be biological replicates, otherwise the error bars just represent the authors pipetting skills.
Response: We agree with reviewer’s comments. Three technical replicates in qPCR has been changed to three biological replicates (line 164-165).

The technical replicates have been changed to biological replicates only in the text.
Figure 6 (previous 7) has not been updated, it is the same as before. If the data shows biological data now, I would expect them to look different than the technical replicates. It does not appear that the authors repeated this experiment with biological replicates, but rather just changed the text and not the data.


Validity of the findings
3) The validation of the RNA-seq data using qRT-PCR (lines 238-245 and figure 7)
The qPCR should use biological replicates instead of technical. The authors conclude that "expression levels ... were all consistent between" qPCR and RNA-Seq, but this is only true for 6 out of the 9 genes analyzed by qPCR.
Response: We agree that the qPCR should use biological replicates instead of technical. Then the expression levels of CL518.Contig3 (PAL), CL2444.Contig2 (C4H), CL1520.Contig3 (4CL), CL3338.Contig1 (CHS), Unigene9945 (CHI), Unigene72602 (F6H), CL2625.Contig2 (IFS2), CL10538.Contig1 (HIDH), Unigene54862 (IF7GT) were all consistent between the qRT-PCR analysis and RNA-Seq data in the article (figure 7 changed to figure 6).

Since the figure has not changed from figure 7 in last version, the comment above about the consistency between qPCR data and RNA-Seq data still holds true. For 6.A, E and F the qPCR and RNA-Seq data are not consistent.

Experimental design

-

Validity of the findings

-

Additional comments

-

---

## Round 0.3 · Major Revisions

Unfortunately, the remaining reviewer was not satisfied with the update of the figure. It does not appear that a new experiment with a biological replicate was done appropriately. Therefore, I would like to request you to present evidence that you re-ran the experiment (or at least far greater clarity/explanation of what and how you did it) and re-revise the manuscript accordingly.

---

## Round 0.4 · Major Revisions

Although the remaining reviewer declares that there are no further points, our section Editor gives the following comments:

"The manuscript reports very detailed findings, but only does so mainly as figures and generic statistics of the findings. There needs to be available data which can be represented in a form for validation for the reader. In the same light the assembled unigenes are discussed without having evidence that they exist for validation. A FASTA set of sequences need be provided with accompanying file linking them to DEG groupings (as presented in Fig 3) and resolvable annotations (such as with GO:123456 designations) to allow easier navigation of the data (as presented in Fig. 8). Some of this is close in the S8 table, but it only refers to closest reference matches, not the actual assembled unigene or contig; this should be provided, deposited in a database repository with resolvable links and annotations. Depending on the quality, NCBI does host some assembled data; that should be looked into. The manuscript is described well, but lacks supporting and usable data; therefore I would recommend further revision to address the transparency of data issues. Some of the presented tables may be modified, or additional supporting data added. Not acceptable at this point."

Could you re-revise the manuscript following the above comments? Thanks for your patience, in advance.

Reviewer 2 ·

Basic reporting

-

Experimental design

-

Validity of the findings

-

Additional comments

Thank you for fixing the issues with the qPCR figure. This addresses the final point I had, and I have no further points.

---

## Round 0.5 · accepted · Accept

I let the section editor confirm the raised points. At my level, the transparency issue has been solved.